# ALP-Based Biosensors Employing Electrodes Modified with Carbon Nanomaterials for Pesticides Detection

**DOI:** 10.3390/molecules28041532

**Published:** 2023-02-05

**Authors:** Stefano Gianvittorio, Isacco Gualandi, Domenica Tonelli

**Affiliations:** Department of Industrial Chemistry “Toso Montanari”, Alma Mater Studiorum-University of Bologna, Viale del Risorgimento 4, 40136 Bologna, Italy

**Keywords:** pesticides detection, electrochemical biosensors, alkaline phosphatase, carbon nanomaterials, enzymatic inhibition

## Abstract

Due to the growing presence of pesticides in the environment and in food, the concern of their impact on human health is increasing. Therefore, the development of fast and reliable detection methods is needed. Enzymatic inhibition-based biosensors represent a good alternative for replacing the more complicated and time-consuming traditional methods (chromatography, spectrophotometry, etc.). This paper describes the development of an electrochemical biosensor exploiting alkaline phosphatase as the biological recognition element and a chemically modified glassy carbon electrode as the transducer. The biosensor was prepared modifying the GCE surface by a mixture of Multi-Walled-Carbon-Nanotubes (MWCNTs) and Electrochemically-Reduced-Graphene-Oxide (ERGO) followed by the immobilization of the enzyme by cross-linking with bovine serum albumin and glutaraldehyde. The inhibition of the biosensor response caused by pesticides was established using 2-phospho-L-ascorbic acid as the enzymatic substrate, whose dephosphorylation reaction produces ascorbic acid (AA). The MWCNTs/ERGO mixture shows a synergic effect in terms of increased sensitivity and decreased overpotential for AA oxidation. The response of the biosensor to the herbicide 2,4-dichloro-phenoxy-acetic-acid was evaluated and resulted in the concentration range 0.04–24 nM, with a limit of the detection of 16 pM. The determination of other pesticides was also achieved. The re-usability of the electrode was demonstrated by performing a washing procedure.

## 1. Introduction

In 1950, the 2,4-dichloro-phenoxy-acetic-acid (2,4-D) herbicide became commercially available, and since then its importance for the residential and agricultural market has been increasing exponentially. 2,4-D is a synthetic auxin that gets absorbed by weeds through their leaves or roots. It belongs to a class of selective hormones which induce an uncontrollable growth and finally cause death to the weeds [1], but without attacking the native vegetation [2]. Currently, two thirds of 2,4-D is used in the agriculture field and one third in the residential field [3]. The main reason why the herbicide is so widely spread is related to the phenomenon called ‘2,4-D drift’ [4], which occurs because the herbicide is highly soluble and volatile [2]. If the agricultural usage of 2,4-D keeps grows in the next decades, the concentration of the herbicide in the environment will increase. Therefore, even if 2,4-D is considered safe today, the assessment could change in the near future [5].

Food and water containing high concentrations of 2,4-D could cause a severe impact on human health. Exposures to high doses of the herbicide have proven to cause a degeneration of the central nervous system, decreased nerve conduction, delayed muscle contraction, etc. [6]. It is also assumed that 2,4-D has a slightly toxic and carcinogenic character [7], but further research on the clinical effects of the herbicide is necessary. The presence of the herbicide above cited as well as other pesticides in water and food will probably attract a lot of media attention in the next years, because of their increasing concentrations. Therefore, the development of rapid, cost-effective, and reliable detection methods is needed [8].

Chromatographic (HPLC and GC) and spectroscopic methods (NMR, mass spectroscopy, etc.) are the most used methods for pesticides detection. They have been employed so far but suffer from different drawbacks, such as time consuming procedures and the requirement for trained personnel [9].

Inhibition-based biosensors represent a good solution for replacing traditional detection methods as these devices are capable of performing in site detection of pesticides with quite high sensitivity and specificity [10]. They employ a biological recognition element, usually an enzyme, that can be inhibited by the target pesticide, leading to a variation in the analytical signal measured at the transducer. The measured signal can be electrochemical (electrochemical biosensors) or optical (luminescent biosensors). Electrochemical biosensors have been extensively adopted for the detection of pesticides, owing to their greater sensitivity, fast fabrication, low costs, portability, and possibility of miniaturization [11,12], which allow analysts to perform measurements using microvolumes of samples [13]. A significant advantage of electrochemical biosensors over luminescent ones is given by the possibility of performing measurements on turbid or colored samples without interferences [11]. 

Among commonly used enzymes for the construction of inhibition-based biosensors there are acetylcholinesterase (Ache) and alkaline phosphate (ALP) [13,14]. ALP-based biosensors are a diagnostic tool capable of measuring concentrations of 2,4-D or other pesticides present in food, water, and other kinds of samples [15]. ALP is a metalloenzyme that contains Mg^2+^ and Zn^2+^ ions in its active centre and catalyzes the dephosphorylation reaction of substrates showing its maximum activity at an alkaline pH and at a temperature of 37 °C [16]. The enzyme is also characterized by a broad substrate specificity and displays a wide variety of possible inhibitors, such as pesticides and heavy metals [15,17]. 

Most of the developed ALP biosensors are electrochemical and are usually fabricated by a chemical modification of a conductive substrate used as a transducer [16,17]. Among electrode materials, glassy carbon is chemically and electrochemically inert, mechanically resistant, and possesses a relatively reproducible surface [18]. Because of these features, glassy carbon electrodes (GCEs) are suitable platforms for the development of electrochemical biosensors [19]. GCE’s surface can be chemically modified with carbonaceous nanomaterials or metal nanoparticles in order to increase sensitivity and biocompatibility, and decrease the overpotentials at which analytes can be detected [20]. Nanomaterials-based biosensors are becoming more and more important in analytical chemistry due to the unique optical and electrochemical properties of nanomaterials, which make the devices novel alternatives to conventional systems, improving the sensitivity, the robustness, and the performance of existing techniques [21,22].

Multi-Walled-Carbon Nanotubes (MWCNTs) are commonly used as electrode modifiers because of their high surface area, electrical conductivity, and biocompatibility [20]. MWCNTs are formed by sheets of graphene folded on top of each other in the shape of a cylinder. The nanotubes thus formed have a diameter greater than or equal to 100 nm and a length of a few μm [20]. Recently, N-doped carbon nanotubes have also been successfully used in the development of biosensors [23] and Single-Walled-Carbon-Nanotubes have been employed to improve the crystallinity of a ferroelectric polymer used as a sensing layer, thanks to their ability to favor the rearrangement of the polymeric lamella crystal [24]. Furthermore, graphene (G) is a nanomaterial that lately has received much attention due to its excellent mechanical, chemical, and electrochemical properties [25]. Graphene is obtained by graphite exfoliation and it is constituted by a single 2-dimensional (2D) layer of a hexagonal structure consisting of sp^2^ hybridized carbon atoms [25]. The large use of graphene is hindered by the difficulty in the preparation of stable aqueous suspensions due to its high hydrophobicity [26]. Furthermore, another drawback of graphene is the fact that its sheets are not thermodynamically stable and over time they tend to reaggregate to reform the starting graphitic structure, especially when they are employed as an electrode modifier. The stacking of graphene sheets leads to a decrease in the electroactive surface and, therefore, of the electrochemical performances [27]. The oxidized form of graphene is known as graphene oxide (GO) and has a higher polarity and lower conductivity than graphene, its aqueous suspensions are easily prepared and display good stability [28]. GO can be reduced again chemically (RGO) or electrochemically (ERGO) in order to partially restore G conductivity [29]. Both MWCNTs and ERGO show good performance as electrode modifiers [29,30].

In 2017, our group carried out an extensive research work on the improvement of the performances of GCEs when modified by MWCNTs/ERGO hybrids compared with the use of a single nanomaterial, highlighting a synergic effect, especially for the detection of phenolic compounds [27]. In that paper a thorough characterization of the hybrid material was also performed in terms of electrochemically active areas, morphology, spectroscopic features, and interaction with the GC support. In MWCNTs/ERGO systems, graphene aggregation is inhibited by carbon nanotubes, which intercalate among graphene sheets, slowing down their aggregation and keeping the surface area of the system high. This effect, however, manifests itself only within a range of MWCNTs/ERGO composition. It is, therefore, important to define the proper mass ratio between the nanocomponents in order to obtain the best possible effect [27]. 

In an attempt to establish a direct electron transfer between the active site of laccase and tyrosinase redox enzymes and the electrode surface and to help the immobilization of the enzyme, equal weight amounts of GO and MWCNts were cast on the GCE, followed by GO electrochemical reduction and by the deposition of the biofilm containing the enzyme [31]. Due to the excellent behavior of the developed biosensors in terms of stability, sensitivity, and reliability, they were successfully employed for determining the polyphenol content of commercial fruit juices.

Based on these studies, we tried to demonstrate the potential of a MWCNTs/ERGO/GCE to support ALP and detect L-ascorbic acid (AA) with the aim to develop an inhibition-based biosensor for the sensitive determination of 2,4-D and other pesticides. A schematic representation of the device working principle is shown in Figure 1.

As shown, the biosensor uses 2-phospho-L-ascorbic acid (AAP) as the enzymatic substrate, which generates L-ascorbic acid as the dephosphorylation product. AA can be electrochemically detected by oxidation at the MWCNTs/ERGO/GCE, generating an oxidation current (I_0_) which is proportional to its concentration. When the ALP gets inhibited by pesticides, the concentration of AA generated by the enzymatic reaction decreases, leading to a decrement of the oxidation current (I_x_) recorded at the electrode. The diminution of AA oxidation current is proportional to the pesticide (inhibitor) concentration and can be exploited for performing pesticides detection.

## 2. Results and Discussion

### 2.1. Fabrication and Optical Charachterization of the ALP-Biosensor

The MWCNTs/ERGO/GCE was built following the procedure illustrated in Figure 2a. 

A MWCNTs/GO suspension 1:1 *w/w* was drop-cast on the GCE surface and the GO → ERGO electrochemical reduction was carried out using the CV technique. The GO reduction takes place at about −1.4 V vs. SCE, and the recorded current decreases from the first to the fifth scan. After the GCE modification, the biosensor was fabricated by drop-casting the ALP enzyme and performing its immobilization by cross-linking with a Bovine Serum Albumin (BSA)/Glutaraldehyde (GA) 1:1 *v/v* mixture. 

Figure 2b illustrates optical microscope pictures related to each step of the fabrication process of the biosensor.

To summarize, the biosensor comprises two layers deposited on the electrode. The layer closest to the GC surface is made of MWCNTs/ERGO and is aimed to promote AA oxidation. An ALP enzyme is deposited onto this layer, and it is immobilized with the cross-linking agents so as to obtain the biofilm. The role of the carbon nanomaterials layer and of the biofilms is reported in the following sections. 

In particular, in Section 2.2 the response of the first layer (MWCNTs/ERGO) towards AA is described; the response of the biosensor toward AAP is reported in Section 2.3; finally, the detection of 2,4-D and other pesticides, based on the inhibition of the enzymatic activity, is described in Section 2.4. 

### 2.2. CM-GCE Response to L-Ascorbic Acid

The response of the GCE electrode modified with a mixture of MWCNTs/ERGO 1:1 *w/w* was evaluated in the presence and in the absence of different AA concentrations inside the three-electrode cell by Cyclic Voltammetry (CV). The same procedure was repeated using a bare GCE electrode. The data obtained using the two kinds of electrodes were compared by evaluating the AA oxidation peak potential (E_p_) and the sensitivity of the current response. Figure 3 shows the CVs recorded at the bare GCE and MWCNTs/ERGO/GCE in 0.1 M TRIS-HCl buffered solution (pH 8.0) without (a) and with (b) 1.0 mM AA. When the electrolyte contains AA, a redox wave is present in both responses. The figure highlights that the modification of the GCE with a MWCNTs/ERGO 1:1 mixture causes a decrease in the AA oxidation peak potential of about 160 mV with respect to the bare GCE (from 0.48 to 0.32 V). The peak currents (I_p_) values linearly depend on the AA concentration (C_AA_) between 0 and 5 mM for both the kinds of electrodes. Figure 3c displays the comparison between the two calibration curves obtained by plotting AA I_p_ as a function of C_AA_. Sensitivities were calculated by dividing the slope of each curve by the geometrical area of the electrode, and are similar, even if the value exhibited by the MWCNTs/ERGO/GCE is slightly higher than the one of the bare GCE.

This evidence confirms the positive synergic effect of the MWCNTs/ERGO mixture toward ascorbic acid oxidation, which causes a notable decrease in the oxidation overpotential as the main effect. The lower AA oxidation overpotential at the MWCNTs/ERGO/GCE surface limits the occurrence of the undesired oxidation of other species that could be present in samples to be analyzed and the possibility of the enzyme denaturation by oxidation. These features make MWCNTs/ERGO/GCE a good platform for realizing enzymatic biosensors.

### 2.3. Biosensor Response to 2-Phospho-L-Ascorbic Acid

To investigate the influence of the concentration of the enzymatic substrate AAP on the electrical signal, the response of the biosensor was evaluated in the presence of different AAP concentrations inside the three-electrode cell and the voltammograms were recorded. Measurements were performed in a pH 8.0 buffered solution at 37 °C, using MgCl_2_ as a cofactor. By overlaying the recorded voltammograms Figure 4a was obtained.

As can be seen, the signal related to AA oxidation (E_p_ 0.3 V vs. SCE) increases by moving to higher AAP (substrate) concentrations, as the enzymatic reaction rate increases, according to the Michaelis–Menten law. This leads to an increase in the AA concentration generated by the enzymatic reaction and, therefore, to an increment of AA oxidation current recorded at the MWCNTs/ERGO/GCE. The enzymatic reaction in which ALP and AAP are involved is shown in Figure 5.

Figure 4b shows the calibration curve obtained by plotting I_p_ as a function of the AAP concentration (C_AAP_). It shows that the sensitivity of the biosensor is almost five times lower than the sensitivity of the MWCNTs/ERGO/GCE shown in Figure 3c. This occurs because of the diffusional barrier caused by the enzymatic layer on the MWCNTs/ERGO/GCE, which reduces the diffusion rate of AA towards the electrode surface [32]. Furthermore, some alterations of ALP active sites, due to the cross-linking immobilization, can also occur so reducing the concentration of the produced AA [33]. The relevant calibration curve in Figure 4b has a correlation coefficient of 0.998, which means that the relationship is linear in the investigated concentration range and, theoretically, any AAP concentration could be chosen for the further electrochemical measurements.

### 2.4. Biosensor Response to the 2,4-D Herbicide

It is well known that ALP is inhibited by 2,4 D, and the interaction results in a high inhibition, which is known to be reversible and of a non-competitive character [16,17,34]. 

The pesticide concentration can be measured from the I_p_ decrease, related to the enzyme deactivation, which leads to a decrement of the AA concentration at the electrode surface. Measuring the current before and after the addition of a certain amount of the 2,4-D inhibitor, the Inhibition Degree (ID) has been calculated for each added 2,4-D concentration, employing the Equation (1):(1)$$\mathrm{ID}=\frac{I_{0}-I_{x}}{I_{0}}$$

$I_{0}$ represents the current measured before 2,4-D addition and $I_{x}$ represents the current measured when a certain 2,4-D concentration is added.

CV was chosen as the electrochemical technique and inhibition measurements were performed by adding increasing 2,4-D aliquots inside the three-electrode cell containing a fixed concentration of AAP. Ten min was chosen as the incubation time between the biosensor and the 2,4-D. 

Chronoamperometry was also tested as a measurement technique, keeping the potential at 0.3 V vs. SCE and performing 2,4-D additions in a stirred solution.

In the resulting chronoamperogram, no clear current decreases were pointed out following the 2,4-D injections due to the decreasing concentration of the generated AA. Furthermore, the current values measured by chronoamperometry were lower in respect to the ones measured by CV. This result can be ascribed to the maintenance of a constant potential over time, which can cause the re-oxidation of graphene on the surface of the MWCNTs/ERGO/GCE, leading to a decrease in the detected oxidation current relative to the AA produced by the enzymatic reaction. In addition, keeping the solution under stirring a significant experimental noise was evident. 

The AAP concentration was chosen in order to achieve the highest inhibition degree possible and a high rate of the enzymatic activity [22].

Table 1 summarizes the results in terms of the 2,4-D inhibition degree obtained with three different AAP concentrations.

In our case, the best response, in terms of 2,4-D inhibition degree, was obtained with the addition of 3.0 mM AAP inside the cell. With the other tested concentrations (5 and 10 mM) the results achieved were significantly worse; therefore, this is the reason why this concentration was employed throughout this work.

Figure 6a shows voltammograms recorded after the addition of different 2,4-D aliquots to the buffered solution containing 3.0 mM AAP, while Figure 6b shows a calibration curve obtained by plotting the ID against the logarithm of 2,4-D molar concentration (C_2,4-D_).

As can be seen from Figure 6a, in the absence of 2,4 D the highest I_p_ is observed. When the signal is recorded in a solution containing 2,4 D, I_p_ decreases from its highest value due to the ALP inhibition. The inhibition degree increases linearly with the logarithm of 2,4-D concentration (Figure 6b).

### 2.5. Reproducibility of the Biosensor Response

To evaluate the reproducibility of the biosensor, the CV measurements, described in Section 2.4., were repeated three times using three different biosensors and performing increasing additions of a 0.1 µM 2,4-D working solution. The relevant analytical parameters are reported in Table 2.

The reproducibility of the biosensor response towards 2,4-D was calculated as the % Relative Standard Deviation (RSD), associated to the mean of the three measurements, and resulted equal to 11% and 0.12 ID/log(C_2,4-D_), respectively. 

The limit of detection (LoD) was calculated from the negative intercept of each calibration curve with the x axis and, averagely, was about 16 pM, even if the three data resulting from the different sensors displayed a significant variability, being from about 0.1 to 40 pM. This variability can be attributed to many reasons, e.g., difficulty in the preparation of the electrode modified with the three layers since the drop casting is performed manually [35], and the very small unit size (10^−12^–10^−9^ M) of the 2,4-D additions.

Pesticide limits in European drinking water are regulated by the Drinking Water Directive (98/83/EC) which establishes a limit of 0.5 µg/L for the pesticides total concentration and a limit of 0.1 µg/L if a single pesticide is present [36]. In the case of 2,4-D the limit is equivalent to a concentration of 452 pM. The average LoD of the developed biosensor is about 30 times lower, and therefore, our result can be considered a good achievement. 

Our results for the detection of 2,4-D were compared with those obtained with similar ALP-inhibition based biosensors, found in the literature. The AA oxidation potential, linear range, and LoD were taken into account for each biosensor and are shown in Table 3. 

Compared with the other biosensors, our device shows the lowest AA oxidation potential and the lowest LoD value. Due to these features, the biosensor displays a better selectivity towards oxidizable interferents, which could be present in the real samples to be analyzed, and it is suitable for the detection of pesticides when their concentrations are very low. As we were interested in detecting low levels of pesticides contamination, the investigated linear range was about two decades lower than the other ranges reported in the Table. Furthermore, our biosensor has proven to be advantageous in terms of operational simplicity, easier fabrication, and low costs. 

### 2.6. Re-Usability of the Biosensor

As already mentioned in the Section 2.4, the literature states that the interaction between ALP and 2,4-D is reversible [16,17,34] and, therefore, the biosensor could be re-used with a suitable washing step aimed at removing the non-covalent interactions between the enzyme and the inhibitor. This treatment can be performed with a non-denaturizing surfactant present in a buffered solution [38]. Three surfactants were tested at low concentration, as it is well known that biological compounds, such as enzymes, are quite sensitive to external influences. The compounds investigated were an anionic surfactant, i.e., Sodium dodecyl sulphate (SDS), and two non -ionic ones, i.e., Tween 80 (Tw80) and Triton X-100 (Tx-100). 

Re-usability measurements were executed by exposing the biosensor to a 2,4-D certain concentration in the presence of AAP and then performing a washing step for 30 min with a 0.1% *v*/*v* surfactant buffered solution (pH 8.0). After the washing step, a new electrochemical measurement in the presence of AAP was carried out and then another addition of 2,4-D was made, to verify if the pesticide is again able to inhibit ALP. Before the use, the biosensor was hydrated for 20 min in a pH 8.0 buffered solution, and an incubation time between ALP and 2,4-D of 10 min was chosen. The best results were achieved with a 0.1% Tx-100 surfactant solution as washing agent and are shown in Figure 7.

The almost identical current peaks before (0 µM 2,4-D (1)) and after (0 µM 2,4-D (2)) the washing step, as well as the inhibition extent displayed after the addition of the same amount of 2,4-D, confirm the re-usability of the biosensor after being in contact with a Tx-100 solution. This evidence reveals the ability of such a surfactant of breaking the interactions between ALP and 2,4-D and proves the reversibility of the inhibition process.

### 2.7. Other Detectable Pesticides

Since the ALP enzyme possesses a wide variety of inhibitors, the detection of other pesticides has been verified. To test the multifunctionality of the ALP biosensor, Malathion, Parathion, and Parathion-methyl were investigated. The experimental conditions were the same as used in the 2,4-D detection. 

In Table 4 are reported the analytical parameters relevant to the quantitative determination of the other pesticides taken into consideration, in the same concentration range of 2,4-D.

Also in this case, the linearity of the response was good, and the LoD values were in the unit size of pM, which is again a good result. From this experimental evidence, we can state that the proposed ALP-inhibition-based electrochemical biosensor appears to be a promising tool for the detection of Malathion, Parathion, Parathion-methyl and, of course, of 2,4-D.

## 3. Materials and Methods

### 3.1. Chemicals

Hydrogen chloride (37%, *w/w*), nitric acid (68% *w/w*), lithium perchlorate, magnesium chloride hexahydrate, L-ascorbic acid (AA), 2-phospho-L-ascorbic acid trisodium (AAP), sodium dodecyl sulphate (SDS), Tween 80, Triton X-100, bovine serum albumin (BSA), glutaraldehyde (GA) (25%, *w/w*) were of reagent grade and were purchased from Sigma-Aldrich. Alkaline phosphatase (ALP) from bovine intestinal mucosa, 37 units/mg solid, was obtained from Sigma Aldrich (Saint Louis, MO, USA). Tris-hydroxymethyl-aminomethane (TRIS) was purchased from VWR chemicals.

Graphene oxide (GO; 4 mg/mL water suspension) and multi-walled carbon nanotubes (MWCNTs; diameter 20–50 nm) were bought from Sigma-Aldrich.

### 3.2. Equipment

All electrochemical measurements were performed by an Autolab potentiostat/galvanostat (Metrohm, Herisau, Switzerland) interfaced to a personal computer and controlled by its dedicated software (GPES 4.9.007). For the processing of the data into graphics, the software OriginPro 9.0 was used. The Autolab potentiostat/galvanostat was coupled to a three-electrode system, consisting out of a Saturated Calomelane Electrode (SCE) as the reference electrode, a working electrode and a platinum wire as the counter-electrode. For the optimal enzymatic activity, the TRIS-HCl buffer pH 8.0 in the cell was heated up to 37 °C by using a heating immersion circulator (Julabo 5 thermostat, Seelbach, Germany). The pH measurement of the TRIS-HCl buffer was carried out with an AMEL 338 pH meter (Milan, Italy). Optical microscope pictures (Figure 2b) were taken using a Dino-Lite digital microscope (AnMo Electronics Corporation, Taipei, Taiwan).

### 3.3. Preparation of the MWCNTs/ERGO/GCE Electrode

The preparation of the modified GCE electrode (MWCNTs/ERGO/GCE) started with the polishing of a GCE electrode surface (3 mm diameter; 7.06 mm^2^ area) by drawing the figure eight on sandpaper 4000 grid (REMET). After the polishing procedure, 10 µL of a MWCNTs/GO 1:1 *w/w* (0.2 mg/mL GO + 0.2 mg/mL MWCNTs) suspension were deposited by drop-casting on the GCE surface, followed by a 15 min evaporation step in a 60 °C oven. The MWCNTs had been previously oxidized for 20 min by using a 6 M HNO_3_ solution and a Sonics VibraCell ultrasonic probe. After the drop-casting of the MWCNts/GO suspension, GO was subjected to a reduction step to partially restore the electrical properties of graphene. This reduction step was performed by CV in the (−1.3 V ÷ 0 V) potential interval, inserting the MWCNTs/ERGO/GCE in a cell containing 0.1 M LiClO_4_. The scan rate was selected at 0.05 V/s and 5 scans were carried out. After the GO electrochemical reduction, the CM-GCE was washed with distilled water and dried under air flow.

### 3.4. Preparation of the Biosensor

10 µL of an ALP solution (10 mg/mL) were drop-casted on the MWCNTs/ERGO/GCE surface followed by the drop casting of 10 µL of BSA/GA 1:1 *v/v* (5 µL of 10 mg/mL BSA + 5 µL of 1.25% GA). To dry the components, both the drop-casting procedures were followed by a 15 min exposure to a nitrogen flow. Nitrogen was used to evaporate the solvents, because it is a non-reactive gas in contrast to air, which contains oxygen and water vapour.

### 3.5. Electrochemical Measurements

All electrochemical measurements made with the biosensor were performed by CV in the (−0.7 ÷ +0.7 V) potential interval, using a 0.05 V/s scan rate and performing 5 scans. Electrochemical measurements made with the bare GCE (Figure 3a-black line) were carried out by CV in the (−0.25 ÷ +0.7 V) potential interval using a 0.05 V/s scan rate and performing 5 scans. Electrochemical measurements made with the MWCNTs/ERGO/GCE (Figure 3a-red line) were performed by CV in the (−0.4 ÷ +0.5 V) potential interval using a 0.05 V/s scan rate and performing 5 scans.

## 4. Conclusions

To conclude, the fabrication of a MWCNTs/ERGO/GCE alkaline phosphatase inhibition-based biosensor was successfully achieved with good reproducibility. The MWCNTs/ERGO mixture revealed to be an optimal electrode modifier, showing a synergic effect towards ascorbic acid oxidation, which increases the sensitivity and decreases the oxidation overpotential of about 160 mV. The response of the biosensor towards 2,4-D herbicide showed a high sensitivity (0.12 ID/decade) and a LoD about 30 times lower (16 pM) than the 2,4-D herbicide law limit for water samples (452 pM). The re-usability of the biosensor was confirmed by using a Triton X-100 buffered solution as washing agent, and the possibility of using the biosensor for detecting other pesticides, such as Malathion, Parathion, and Parathion-methyl, was also demonstrated. Thanks to this research it will be possible in the future to build the same biosensor on more compact and portable electrochemical devices, such as screen-printed electrodes, inkjet-printed electrodes, and Organic Electrochemical Transistors, in order to realize a miniaturized version of the described ALP-biosensor capable of working with microvolumes of samples.

## Figures and Tables

**Figure 1 molecules-28-01532-f001:**
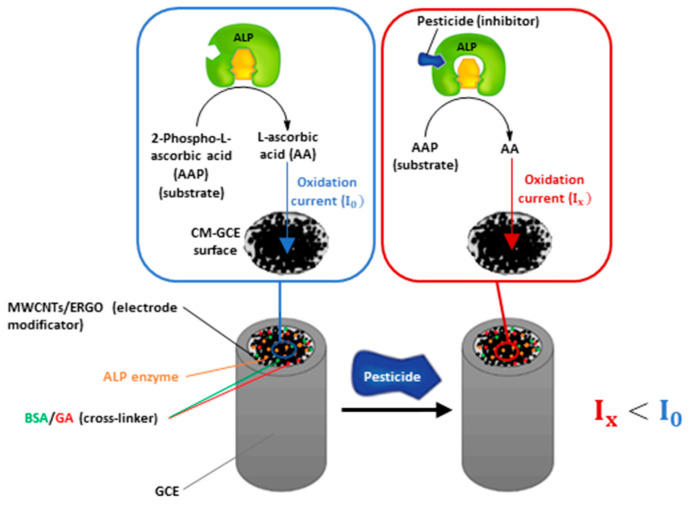
Schematic representation of the biosensor working principle.

**Figure 2 molecules-28-01532-f002:**
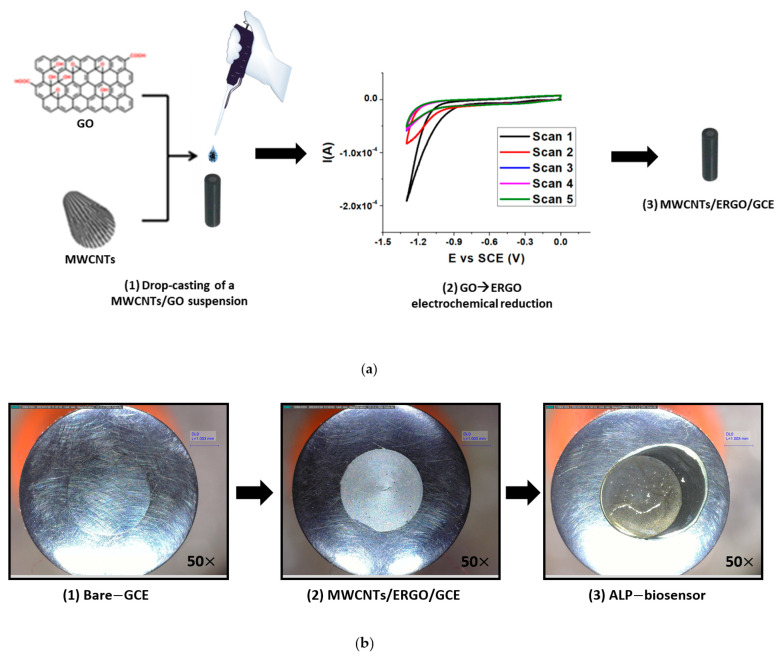
Schematic representation of the fabrication procedure of the MWCNTs/ERGO/GCE (**a**) and optical microscope pictures (50× magnification) related to each step of the fabrication process of the biosensor (**b**).

**Figure 3 molecules-28-01532-f003:**
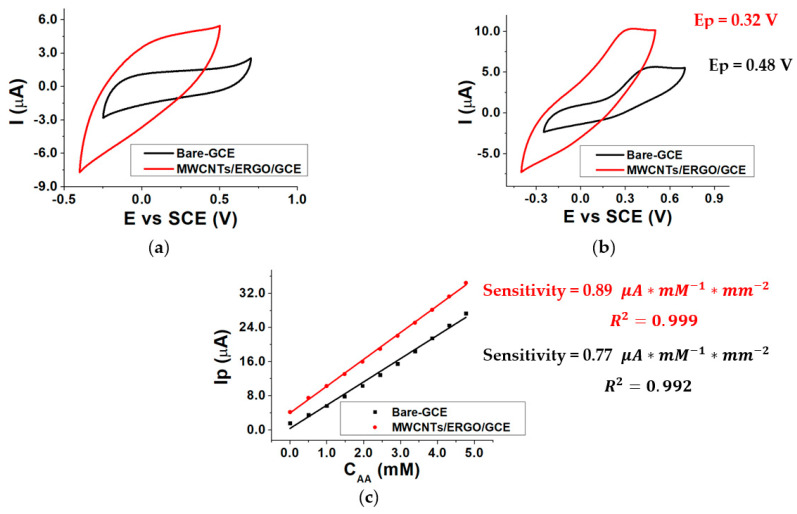
CVs recorded in the absence of AA (**a**)**,** in the presence of 1.0 mM AA (**b**), for bare-GCE and MWCNTs/ERGO/GCE electrodes and calibration curves recorded for Bare-GCE and MWCNTs/ERGO/GCE electrodes (**c**); conditions: 0.1 M TRIS-HCl buffered solution, pH 8.0.

**Figure 4 molecules-28-01532-f004:**
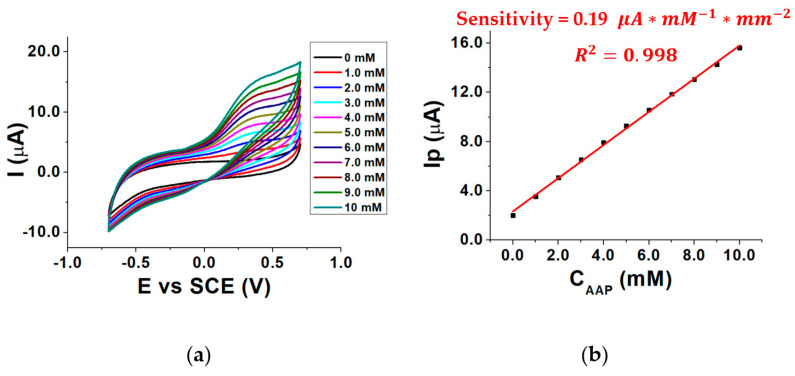
CVs recorded in the presence of 0–10 mM AAP (**a**) calibration curve for AAP (**b**); conditions: 0.1 M TRIS-HCl buffered solution, pH 8.0, T = 37 °C, 0.40 mM MgCl_2_.

**Figure 5 molecules-28-01532-f005:**
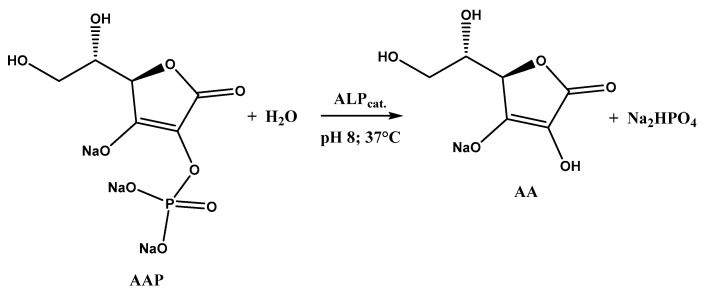
ALP-catalyzed dephosphorylation of AAP.

**Figure 6 molecules-28-01532-f006:**
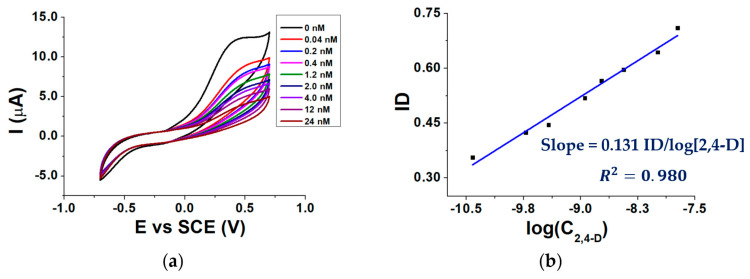
CVs recorded in the presence of 0.04–24 nM 2,4-D (**a**) calibration curve of the biosensor for 2,4-D (**b**); conditions: 0.1 M TRIS-HCl buffered solution pH 8.0, T = 37 °C, 0.40 mM MgCl_2_, 3.0 mM AAP.

**Figure 7 molecules-28-01532-f007:**
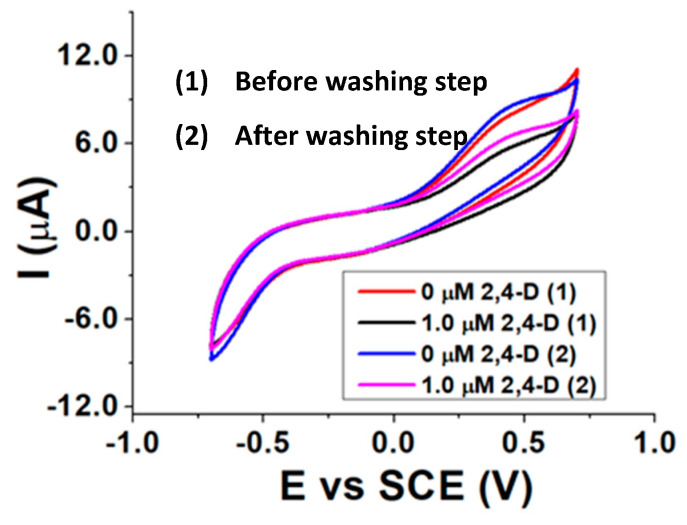
CVs recorded before (1) and after (2) washing the biosensor with a Tx-100 buffered solution (pH 8.0).

**Table 1 molecules-28-01532-t001:** Inhibition degree obtained with different AAP concentrations; conditions: 0.1 M TRIS-HCl buffered solution pH 8.0, T = 37 °C, 0.40 mM MgCl_2_.

AAP Concentration	2,4-D Concentration	Inhibition Degree (ID)
3.0 mM	40 pM	7.68%
5.0 mM	40 pM	2.91%
10 mM	40 pM	3.26%

**Table 2 molecules-28-01532-t002:** Analytical parameters of three different ALP-biosensors toward 2,4-D.

Biosensor	1	2	3
**2,4-D concentration range**	0.04–24 nM	0.04–24 nM	0.04–24 nM
**Equation** **of the calibration curve **	y = 0.11x + 1.1	y = 0.11x + 1.2	y = 0.13x + 1.7
**Correlation coefficient (R^2^)**	0.856	0.925	0.983

**Table 3 molecules-28-01532-t003:** Analytical performances in 2,4-D detection of other ALP-inhibition-based biosensors found in the literature.

Biosensor Structure *	AA Oxidation Potential (V vs. SCE)	Linear Range (nM)	LoD (nM)	Ref.
**Pt/CellAc/GOD-Nylon/ALP**	0.36	23–181	9.0	[16]
**SPE-MWCNTs/PVA-SbQ/ALP**	0.61	9.5–498	4.5	[17]
**SPCE/ALP/sol-gel chitosan**	0.56	4.5–271	2.3	[15]
**SPCE/Fe_2_O_3_/ALP/sol-gel chitosan**	0.56	2.3–136	1.8	[37]
**GCE/MWCNTs-ERGO/ALP/BSA-GA**	0.30	0.04–24	0.02	This work

* List of the abbreviations: CellAc = cellulose acetate, GOD = glucose oxidase, SPE = screen printed electrode, PVA-SbQ = polyvinyl-alcohol with styrylpyridinium groups, SPCE = screen printed carbon electrode.

**Table 4 molecules-28-01532-t004:** Analytical parameters of the ALP-biosensor towards different pesticides.

Pesticide	Malathion	Parathion	Parathion-Methyl
**Substrate**	3.0 mM AAP	3.0 mM AAP	3.0 mM AAP
**Pesticide concentration range**	0.02–14 nM	0.02–14 nM	0.02–14 nM
**Equation** **of the calibration curve**	y = 0.10x + 1.2	y = 0.075x + 0.82	y = 0.096x + 1.1
**Correlation coefficient** (**R^2^**)	0.970	0.981	0.975
*** Limit of detection** (**LoD**)	2.7 pM	9.9 pM	2.1 pM

* LoDs were calculated from the negative intercept of each calibration curve with the x axis; y = ID; x = log (C_Pesticide_).

## Data Availability

The data presented in this article are available on request from the corresponding author. The data are not publicly available yet, due to a two-year embargo on the thesis work in which they are contained.

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
