# Peer review of "ALP-Based Biosensors Employing Electrodes Modified with Carbon Nanomaterials for Pesticides Detection"

_molecules, 2023, doi:10.3390/molecules28041532_

Round 1
Reviewer 1 Report
In this paper, the authors reported “ALP based biosensors employing electrodes modified with carbon nanomaterials for pesticides detection”. In this work, the authors suggested development of an inhibition-based biosensor employs a MWCNTs/ERGO modified GCE as transducer and ALP as biological recognition element. Especially, A focus point of this work is formation of a new type of ALP biosensor using BSA and GA materials and thier application of electrochemical devices. In my opinion, this manuscript would be useful for this journal and near fields. However, in main text, there are unclear points in the main manuscript as the following;
1.In Introduction part, the authors described the background of this research, focusing on biosensing and carbon materials. However, I was unable to read the key points of this study from the above. Probably, I understand that the production of ALP biosensors and their application to sensing systems are important, but it seems that the information around them is lacking. Or is the application of nanocarbon materials to sensors new? If YES, I think the authors should add more information not only “Characteristics of Carbon materials” but also “Sensor application of carbon materials”. In that case, please add the REF of the nanocarbon type sensor as below, for better understanding of the reader;
#1_Y.-X. Wang et al., Carbon, 186, 406 (2022). (N-doped carbon nanotubes for biosensor)
#2_T. Sekine et al., Small Science, 1, 2100002 (2021).(Nano carbons composite materials for sensing devices)
2. In Fig.2, you showed schematic representative illustrations of the fabricated biosensor. However, I cannot image a real sensor device. The authors should add the photos your original sensor device.
3. No problems were found in the data. However, in section of Conclusion, the content of the summary is ambiguous. For example, you mentioned as “we present the development of an inhibition-based biosensor which employs a MWCNTs/ERGO modified GCE (CM-GCE) as transducer and ALP as biological recognition element.” in Introduction, but in Conclusion, you only mentioned as only the characteristics of the fabricated sensor. I recommend you should organize and describe the importance of your research again.
Reviewer 2 Report
The manuscript developed a CM-GCE alkaline phosphatase inhibition-based biosensor to detect pesticides. The design strategy and application are interesting, but some improvements and detail corrections are required before it might be considered for publication in Molecules. For this reason, see the detailed comments presented below.
1. Electrochemical detection of pesticides based on alkaline phosphatase inhibition has been reported before. For example: 10.1016/j.jelechem.2004.08.004, 10.1007/s00216-022-04279-x. What is the advantage of this method compared to previously reports?
2. There are too many paragraphs in the introduction, and the writing lacks logic and transition.
3. The preparation of the electrode, including the functionalization of MWCNTs/ERGO, BSA, GA and ALP, lack necessary characterization and identification.
4. What if MWCNTs or ERGO is used individually instead of together? The synergistic effect of MWCNTs/ERGO should be discussed systematically.
5. What is the interaction between BSA and MWCNTs/ERGO modified electrode?
6. A comparison between the proposed method and previously reported methods including linear range, LOD and so on should be added to prove the superiority of the current ALP based biosensor.
7. Minor suggestion: There is an unexpected bracket in Figure 2c. All figures in the manuscript are suggested to be checked and beautified.
Round 2
Reviewer 1 Report
The authors properly addressed my concerns. I recommend the publication.
Reviewer 2 Report
All questions are addressed well, and the manuscript can be published with the current version.